# Hybrid Receptor-Mediated Molecular Delineations in TNF-α and IGF-1-Induced Costimulatory Effects

**DOI:** 10.3390/ijms262010027

**Published:** 2025-10-15

**Authors:** Chandra S. Boosani, Pradeep N. Subramanyam, Gopal P. Jadhav

**Affiliations:** 1Division of Animal Sciences, College of Agriculture, Food and Natural Resources, University of Missouri, Columbia, MO 65202, USA; pbnk89@missouri.edu; 2University Hospital—MU HealthCare, University of Missouri, Columbia, MO 65212, USA; 3Soma Life Science Solutions, Lewisville, NC 27023, USA; 4Department of Pharmacology & Neuroscience, School of Medicine, Creighton University, Omaha, NE 68178, USA

**Keywords:** TNF-α, IGF-1, IGF1R, TNFR1, SOCS3, inflammation, signaling pathway, tumor suppressor

## Abstract

The growth-promoting roles of IGF-1 (Insulin-like Growth Factor 1) and the inflammation-modulating cytokine TNF-α (Tumor Necrosis Factor-alpha) have been substantially deciphered in various pathological conditions. Also, their biphasic roles in modulating cellular inflammation have been reported. While their independent signaling pathways are sufficiently explored, recent studies have identified that their combined or costimulatory effects uniquely contribute to the regulation of different disease states. Such costimulatory effects appear to confer divergent and at times opposite effects on certain cellular processes. We and others in the literature have previously demonstrated that TNF-α and IGF-1 would independently induce the expression of SOCS3 (Suppressor of Cytokine Signaling-3, a tumor suppressor protein). However, their combined presence has been found to prevent SOCS3 expression. The cause of this divergent opposite effect remained unclear. Here, we provide structural evidence on the interactions between the receptors for TNF-α and IGF-1, and the expression patterns of intermediary proteins which play a prominent role in SOCS3 expression. Our analysis here presents new evidence which demonstrates that TNFR1 (Tumor Necrosis Factor Receptor-1) has the potential to form hybrid receptors with IGF1R (Insulin-like Growth Factor-1 Receptor). Formation of these hybrid receptors may preclude the intracellular signaling that leads to the inhibition of SOCS3. Additionally, we also identified a unique binding site on TNFR1, where SOCS3 by binding to this region is hypothesized to prevent the export of TNFR1 to cell surface. This could be one of negative feedback regulation mechanisms of SOCS3 associated with preventing inflammatory signaling. Our results described here delineate molecular mediators which could cause inhibition of SOCS3 when both TNF-α and IGF-1 induce their costimulatory effects.

## 1. Introduction

Cellular responses to inflammatory stimuli are required to protect and restore the healthy state in cells. Often, it is the sustained and chronic inflammation that leads to disease when not addressed. In several inflammatory conditions, a prominent role of TNF-α in promoting cellular inflammation was reported [1,2]. Besides TNF-α being recognized as a positive acute phase reactant and a pro-inflammatory cytokine, it has also been shown to exert anti-inflammatory responses, which demonstrates its biphasic properties. Similarly, the growth factor IGF-1 has been found to confer both pro- and anti-inflammatory effects in cells. These two ligands have been reported to induce production of different cytokines which amplify inflammatory pathways, and also inter-regulate the expression of each other [3,4,5,6,7]. On the same note, the molecular pathways independently induced by TNF-α and IGF-1 in healthy and disease conditions have been adequately explored. While much of the signaling induced by IGF-1 is mediated through its receptor IGF1R, in case of TNF-α, its signaling has been found to be mediated through its receptors TNFR1 and TNFR2 (Tumor Necrosis Factor Receptor-2). However, ample evidence in the literature suggests that TNFR1 is the major receptor for TNF-α-mediated signaling, and moreover, the signaling mechanisms induced by TNFR2 cross-talks with TNFR1 pathway [8].

We and others in the literature have previously reported that TNF-α and IGF-1 independently induce the expression of tumor suppressor protein, SOCS3 [9,10,11]. This induced expression of SOCS3 is primarily seen as a response to the production of different cytokines. Such a response can be anticipated since SOCS3 functions as a negative feedback inhibitor, and this can also be seen as a defense mechanism required to restore normalcy. However, when both TNF-α and IGF-1 are present, their combined presence renders the opposite effect, that leads to SOCS3 inhibition. These unique effects on the cellular processes induced from the costimulatory effects of TNF-α and IGF-1 have been reported by us and others in the field [10,11,12,13,14]. However, the molecular mechanisms that lead to SOCS3 inhibition when both TNF-α and IGF-1 are present, remained unclear.

Unlike TNF-α, the IGF-1-induced pathways involves signaling that is predominantly mediated through IGF1R. One of the notable features of IGF1R is its ability to interact with other receptors on the cell surface and initiate divergent molecular mechanisms. It was previously shown that IGF1R interacts with thyrotropin and insulin receptors on the cell surface, and forms hybrid receptors with them [15,16,17,18]. On a similar note, TNFR1 was reported to interact with other members within its super family of proteins such as BAFF (B-cell Activating Factor), suggesting that TNFR1 has the ability to participate in hybrid receptor formation [19,20,21,22,23]. In the present work, we show evidence that IGF1R structurally interacts with TNFR1, and this interaction could be responsible for the costimulatory effects induced by TNF-α and IGF-1. Furthermore, by analyzing the microarray data, we identified intermediary proteins involved in the TNF-α- and IGF-1-induced pathways which may be responsible for the inhibition of SOCS3. In addition, we also show that SOCS3, by binding to a unique sequence on TNFR1, prevents its export to the cell surface, which could be part of the feedback inhibitory properties of SOCS3 required for modulating cellular inflammation.

## 2. Results

### 2.1. Gene Expression Analysis from Microarray 

Previously, we showed unique effects of TNF-α and IGF-1 co-stimulation on cells [10,11,13,14,24,25]. A recent study similar to our previous reports, conducted microarray based comprehensive analysis using BeWo human trophoblastic cells by treating them with TNF-α or IGF-1, or with both TNF-α and IGF-1 together [26,27]. The study used total RNA from duplicate samples that were hybridized onto Agilent-072363 SurePrint G3 Human GE v3 8 × 60 K Microarray (Aligent Technologies, Sanat Clara, CA, USA) and generated GSE119651 database [26,27]. Subsequently, using the NCBI GEO2R embedded tool, transcript profiles of TNFR1, TNFR2, IGF1R, IGFPB3, DCN, SRC, SRCIN1, and SOCS3 were retrieved from GSE119651 database. The raw data was analyzed in Excel and presented in Figure 1. Only the groups that were treated with 100 ng/mL of TNF-α, with or without 100 ng/mL of IGF-1 were used in our analysis. Based on the information from the above microarray study, we conducted protein network analysis to identify molecular mediators that regulate TNF-α- and IGF-1-mediated pathways (Figure 2).

### 2.2. Assessment of Protein–Protein Structural Interactions

To understand TNF-α- and IGF-1-induced costimulatory effects on SOCS3, protein structural interactions between TNF-α, IGF-1, SOCS3, IGF1R, and TNFR1 were investigated. Molecular docking was performed on the ClusPro webserver, by submitting the pdb codes for the corresponding proteins along with their chain specifications. After docking, the interactions between the proteins were examined using PyMol software, version 2.6.0. The number of polar contacts that represent hydrogen bonding between the chains were counted manually. Table 1 shows a selected list of protein–protein interactions assessed. The number of polar contacts observed between the two interacting proteins, along with the number of members participating in the cluster interacting, and their weighted energy scores are shown.

### 2.3. SOCS3 Interactions with TNF-α Receptor

Protein–protein docking analysis was performed to evaluate the interaction between SOCS3 and TNFR1 (Figure 3A), as well as SOCS3 and TNFR2. Comparative analysis revealed that SOCS3 engages in more extensive polar contacts with TNFR1 than with TNFR2, suggesting a higher binding affinity toward TNFR1. Structural modeling demonstrated that the cysteine-rich domain 4 (CRD4) A-loop of TNFR1 inserts into the open cavity formed by flexible SH2-domain residues of SOCS3 (Figure 3B). In contrast, the B-loop of the TNFR1 CRD4 region becomes buried within an additional pocket shaped by the SH2 domain, N-terminal residues, and central β-sheet elements of SOCS3. Specific residues that participate in structural interactions were highlighted in Figure 3.

In addition, Expasy analysis of the proteolytic cleavage site of the SOCS3-binding region on TNFR1 was identified (Figure 4). The amino acid sequence of TNFR1 shows the target cleavage sites for specific proteases that are released by *Staphylococcus aureus* and *Pseudomonas* species, suggesting potential pathogen-driven modulation of SOCS3 mediated regulation of TNFR1.

### 2.4. Analysis of Hybrid Receptor Formation

Structural docking analysis revealed that the interaction between IGF1R and TNFR1 leads to the formation of a heterotrimeric hybrid receptor complex (IGF1R:TNFR1:TNFR1), Figure 5A,B. The observed structurally interacting domains in the heterotrimeric hybrid receptor, agrees with the previously characterized hybrid receptor formation between IGF1R and Insulin receptors [17]. Figure 5C,D highlights specific residues that participate in the formation of hybrid receptors. Notably, the CRD3 domain of TNFR1 established polar contacts with residues of both the L1 and L2 domains of IGF1R, stabilizing the hybrid interface. These interactions suggest that the IGF1R:TNFR1 hybrid complex occupies overlapping ligand-binding interfaces for both IGF-1 and TNF-α, indicating potential modulation of downstream signaling pathways.

### 2.5. Hybrid Receptor Formation Alters IGF-1 Binding Topology and Its Signaling Dynamics

Our molecular docking also identifies unique interactions of IGF-1 with the heterotrimeric hybrid receptor (Figure 6). The structural overlay of IGF-1 ligand within the hybrid receptor complex revealed a conformational shift in its binding orientation compared to native IGF1R interaction. The architecture of the hybrid receptor comprising IGF1R domains such as L1 (wheat tint), and L2 (orange) interacting with the extracellular domain of TNFR1 (cyan) were shown. IGF-1 binding to the hybrid receptor complex favored association with the dimeric TNFR1 interface, at the CRD4 regions, compared to its canonical binding site at the L1 domain of IGF1R. Spatial displacement of the IGF-1 ligand bound to TNFR1 relative to IGF1R, indicates upward shift of approximately 15.2 Å in the flexible loop and 18.4 Å in the β-sheet region (Figure 6B). These findings suggest that the formation of hybrid receptors may alter the IGF-1 binding topology, potentially modulating receptor activation and downstream signaling dynamics.

## 3. Discussion

The biphasic roles of the cytokine TNF-α, and its receptor TNFR1 that promote and/or prevent the underlying disease conditions are evident in the literature. Also, the mechanisms associated with the regulation of cellular inflammation have been investigated in detail across many pathological conditions [1]. TNF-α, an acute phase reactant, primarily mediates its signaling through TNFR1 and TNFR2. However, TNFR1 was reported to express across many cell types compared to TNFR2. Similar to TNF-α-mediated signaling, the growth factor IGF-1 has also been reported to have a dual role in regulating cellular inflammation [28]. While the molecular pathways individually regulated by TNF-α or IGF-1 were deciphered, we and others in the literature have reported that the presence of both TNF-α and IGF-1 together would induce unique effects in regulating cellular processes [9,10,11,13,14,24,25,29]. Here, we attempt to delineate the molecular mediators that are altered when the two ligands, TNF-α and IGF-1, are present. Besides the regulation of the Jak/Stat pathway by TNF-α and IGF-1, we have previously shown that together these two ligands induce epigenetic changes that regulate DNA methylation and microRNAs that target DNMT1 [11,13,14,24]. One of the target proteins we identified previously whose expression is selectively regulated by TNF-α and IGF-1 is the tumor suppressor protein, SOCS3. While both TNF-α and IGF-1 would induce the expression of SOCS3 when treated independently, co-treatment with both the ligands was found to prevent SOCS3 expression. It is still unclear how this combined presence of TNF-α and IGF-1 induces an opposite effect in preventing the expression of SOCS3. To investigate this aspect, we analyzed data from the microarray to delineate the underlying molecular mechanisms and conducted detailed structural assessments of TNF-α and IGF-1, and their receptors along with SOCS3.

In a recent study, using BeWo human trophoblastic cell line the costimulatory effects of TNF-α and IGF-1 were evaluated through transcriptomic analysis [26,27]. The authors showed that the combined presence of TNF-α and IGF-1 promoted NFκB signaling, arachidonic acid pathways and epigenetic regulations such as CpG methylation. Notably, the study revealed the costimulatory effects of TNF-α and IGF-1 on preeclampsia with increased insulin resistance. In vitro BeWo cell culture and treatment with TNF-α and IGF-1 supports studies on different inflammatory conditions including preeclampsia. By accessing the above microarray database, we analyzed the expression patterns of select genes of interest TNFR1, TNFR2, IGF1R, IGFPB3, DCN, SRC, SRCIN1, and SOCS3 (Figure 1). Our results using the microarray database agree with our previous reports where we demonstrated that in the presence of both TNF-α and IGF-1, SOCS3 expression gets silenced due to increased DNA methylation in the CpG island of SOCS3. This was correlated with elevated levels of DNA methyltransferase DNMT1 [13,14,24]. Based on our observations in Figure 1, we conducted pathway screening using NetworkAnalyst as detailed below. Of note, pathways related to epigenetic regulators and apoptosis were excluded from our current analysis.

We screened the STRING database through NetworkAnalyst, using the known mediators of TNF-α and IGF-1 signaling pathways [30,31,32,33]. In Figure 2, we show some of the important mediators that are common to both TNF-α- and IGF-1-mediated pathways. The schematic representation shows that both TNF-α and IGF-1 have molecular mediators that indirectly converge to regulate SOCS3 expression. It is to be noted that the analysis using NetworkAnalyst is based on the reports that were published thus far; however, it does not detail whether the protein mediators identified in these co-integrated pathways would interact. To address those deficiencies, we carried out molecular docking with ClusPro, and the resulting docked conformations as pdb files were visualized using PyMol to analyze protein–protein interactions. Table 1 shows a select list of pdb structures imported from RCSB database that were used for molecular docking. The energy represented as a weighted score is not the sole criterion for determining structural interactions. Thus, from the top docked plausible interacting coordinates after docking, the model with most stable protein–protein interactions along with lowest weighted scores were used for assessing protein–protein interactions.

In Figure 3, we show SOCS3 interaction with TNFR1. SOCS3 interaction with TNFR2 was also evaluated, however a very low binding efficiency was observed between the two. At the atomic interaction level, multiple stabilizing polar contacts were observed between SOCS3 and TNFR1. Specifically, the amide group of TNFR1 Asn116 and the backbone carbonyl of Cys117 interacted with the backbone carbonyl and amide groups of SOCS3 Gln121. The hydroxyl group of Ser118 in TNFR1 formed polar interactions with SOCS3 Ala124 and Gln125 backbone amide groups, while Leu119 backbone amide established hydrogen bonds with the side-chain amide of Gln125. Furthermore, the backbone amides of Cys120 and Thr124 in TNFR1 established hydrogen bonds with the hydroxyl groups of SOCS3 Ser115 and Thr112. Additional interactions identified include Val125 backbone amide and Thr124 hydroxyl groups from TNFR1 engaging with SOCS3 Thr112 hydroxyl and Pro110 backbone carbonyl groups.

Distinct interactions were also identified in the B-loop CRD4 region of TNFR1. The carbonyl oxygen of Asn120 in TNFR1 formed a polar contact with Lys11 of the SOCS3 N-terminal domain, while the Asn120 backbone amide interacted with Asp89 of the SOCS3 central β-sheet. Within the SH2 domain, the backbone carbonyl of His126 and the backbone amide of Ser128 in TNFR1 interacted with SOCS3 Thr131 backbone amide and Ser130 hydroxyl groups, respectively. Moreover, the backbone carbonyl of His140, the backbone amide of Gly142, and the backbone carbonyl of Phe143 in TNFR1 engaged in polar contacts with the hydroxyl group of SOCS3 Thr104.

Interestingly, the above binding region of SOCS3 was found at the C-terminus region of the ectodomain of TNFR1. This region lies between the last of the three FN3 fibronectin domains and the transmembrane domain. Further analysis of the amino acid sequence of the interacting region between SOCS3 and TNFR1 using Expasy, showed the presence of proteolytic cleavage sites for *Staphylococcus aureus* and *Pseudomonas* species were noted (Figure 4). This SOCS3 binding site on TNFR1 lies in its extracellular domain and this region would be susceptible to the actions of proteases secreted by bacterial species. These protease cleavage sites on TNFR1 may be associated with the receptor shedding of TNFR1 that was reported previously [34]. Since SOCS3 is not secreted and always remains inside the cell, we speculate that SOCS3, by binding to its specific site on the TNFR1, hinders receptor export onto the cell surface and prevents TNF-α–TNFR1-induced signaling. It can be presumed that such interactions may be part of the negative feedback regulation properties of SOCS3.

Similar to TNF-α, the growth factor IGF-1 has also been reported to mediate both pro- and anti-inflammatory signaling. In addition to inducing the production of certain cytokines, IGF-1 was reported to induce the expression of SOCS3 and also TNF-α [24,35]. We first evaluated protein interactions between SOCS3 and IGF1R. In our molecular docking analysis, we did not observe any polar contacts between SOCS3 and IGF1R. Since the costimulatory effects of TNF-α and IGF-1 inhibit SOCS3 expression, and no structural interaction observed between IGF1R and SOCS3, we hypothesized the possibility of hybrid receptor formation between IGF1R and TNFR1. Of note, both TNF-α ligand and the receptor TNFR1 were reported to exist in soluble and also membrane-bound forms [36]. Thus, the interactions between ectodomains of both TNFR1 and IGF1R were evaluated (Figure 5A). In Table 1, we show the list of selected protein interactions that were evaluated. The pdb codes, the number of polar contacts, and the docking score representing lowest energy from the interactions assessed were shown. We observed lowest docking energy and highest number of polar contacts between TNFR1 and IGFR1. These findings suggest that TNFR1 not only plays an important role in regulating SOCS3 expression, but importantly it provides evidence on the formation of hybrid receptors between TNFR1 and IGF1R.

Importantly, as seen in Figure 5A, the formation of hybrid receptors between IGF1R and TNFR1 supports earlier reports which suggest that IGF1R forms hybrid receptors with thyrotropin and insulin receptors (IR) [18,37,38,39,40,41]. It was also reported that treatment with TNF-α leads to reduced sensitivity in trophoblast cells towards insulin and IGF-1 through formation of IGF1R and IR hybrid receptors [29]. The structural interaction between IGF1R and IR hybrid receptors (PDBs: 7S0Q, 7S8V) and also between IGF1R and thyrotropin receptors was previously reported. Our analysis of the heterotrimeric hybrid receptors agrees with the above reports. As shown in Figure 5A,B, we observed that IGF1R structurally interacts only with TNFR1. Herein, only the interacting region of IGF1R was shown. It is generally conceived that the ligand TNF-α exists in a homotrimer state. Receptor clustering of TNFR1 was speculated to be auto-induced and exist in a homotrimer state on the cell surface. Subsequently, the trimeric form of TNF-α interacts with the trimeric TNFR1 cell surface to initiate TNF-α-induced effects [42,43]. Based on our observations and above reports, we speculate that the hybrid receptors would exist in a heterotrimeric state. This may be required not only for receptor trimerization but also for structural stabilization on the cell surface. As seen in Figure 5, our molecular docking predicts the formation of heterotrimeric hybrid receptors as IGF1R:TNFR1:TNFR1. However, we do not exclude the existence or possibility of other polymeric assembly states in hybrid receptor formation, which needs to be determined through crystallographic studies.

Our structural analysis of this hybrid receptor shows that the interaction between TNFR1 and IGF1R occupies the binding site of TNF-α on the TNFR1. Figure 5C,D shows that the Gly445 backbone carbonyl of IGF1R-L2 forms polar contact with Lys79 sidechain amine of TNFR1. Additional polar contacts between Arg99 of TNFR1 and the aromatic hydroxyl groups of Tyr391 and Tyr417, as well as the carboxyl group of Glu385 from IGF1R-L2, were also observed. Structural stabilization can be anticipated from the sidechain interactions between Lys359 and Arg361 of IGF1R-L2 with Gln102, Cys114, and Asn116 of TNFR1, respectively. In Figure 5D, residues within the IGF1R-L1 domain, specifically Arg59 and Arg112, can be seen establishing polar contacts with Thr124 (A-loop) and Glu147 (B-loop) of TNFR1. The hetero-NH group of His30 in IGF1R-L1 engaged in hydrogen bonding with the Gly142 backbone carbonyl of TNFR1, while aromatic π–π stacking was observed between His30 and Tyr128 of IGF1R-L1 as well as between His30 of IGF1R-L1 and Phe143 of the TNFR1 A-loop. Additionally, Leu33 and Ser35 of IGF1R-L1 participate in establishing polar contacts with Ala141 and His140 of the TNFR1 B-loop. These interactions suggest that IGF1R:TNFR1:TNFR1 hybrid complex occupies overlapping ligand-binding interfaces for both IGF-1 and TNF-α, indicating potential modulation of downstream signaling pathways.

Also, the structural overlay of IGF-1 ligand within the hybrid receptor complex revealed a conformational shift in its binding orientation compared to native IGF1R interaction (Figure 6). Figure 6 also shows that IGF-1 can exhibit preferential binding toward the TNFR1 interface rather than the canonical IGF1R-L1 site in the hybrid receptor. The observed displacement of IGF-1’s flexible loop (15.2 Å) and β-sheet (18.4 Å) suggest a conformational change that may alter the receptor’s extracellular topology involving ligand-binding dynamics. Such reorientation entails possible modulation of hybrid receptor activation, potentially affecting TNFR1-mediated inflammatory signaling with IGF1R-driven cellular pathways as discussed above. Therefore, as a result of hybrid receptor formation, SOCS3 expression gets hindered. Formation of the hybrid receptor and subsequent costimulatory effects of TNF-α and IGF-1 would have therapeutic implications in different human diseases and complications. These diseases and complications include Duchenne muscular dystrophy, Preeclampsia, Alzheimer’s disease, lung cancer, and cardiovascular restenosis in which both TNF-α and IGF-1 have been reported to exert their costimulatory effects [5,12,13,44,45].

To summarize, the present study provides evidence on the possibility of a new molecular mechanism that stems from the costimulatory effects of TNF-α and IGF-1 ligands. This study sufficiently addresses the underlying cause for SOCS3 inhibition when both TNF-α and IGF-1 are present. To our understanding, this is the first report that presents evidence on the possibility of formation of heterotrimeric IGF1R:TNFR1:TNFR1 hybrid receptors. In addition, this study also suggests that SOCS3 binds to a unique sequence on TNFR1. This could prevent receptor export to the cell surface; one of the possible negative feedback regulatory mechanisms of SOCS3.

## 4. Materials and Methods

### 4.1. Transcriptomic Analysis

The transcriptomic profiles in GSE119651 microarray database was previously generated using BeWo human trophoblastic cells that were treated with TNF-α or IGF-1, or with TNF-α and IGF-1 together [26,27]. As stated, the samples were run in duplicates by hybridizing the total RNA from control and treated cells onto Agilent-072363 SurePrint G3 Human GE v3 8 × 60K Microarray (Aligent Technologies, Sanat Clara, CA, USA). The expression profiles of select genes TNFR1, TNFR2, IGF1R, IGFBP3 (Insulin-like Growth Factor Binding Protein-3), DCN (Decorin), SRC (Sarcoma Kinase), SRCIN1 (Sarcoma Kinase Inhibitor-1), and SOCS3 were extracted from GSE119651 database and analyzed.

### 4.2. NetworkAnalyst

On the NetworkAnalyst webserver [30,31,32], the option for “Gene list input” was selected and a list of the official gene symbols of 45 human genes that are involved in regulating TNF-α- and IGF-1-mediated signaling was uploaded as input data. The Generic Protein–Protein Interaction with First order build option was used, and the STRING interactome database set at 900 confidence score (allowable range is 400–1000, with 900 default option) was selected. This resulted in a subnetwork with 819 nodes and 45 seeds. Nodes such as the ones representing epigenetic gene regulators, apoptosis, and other quaternary connections showing different isoforms of various proteins that are remotely related to the pathways of interest were excluded.

### 4.3. Protein Docking at ClusPro Webserver

All PDB files listed were acquired from the RCSB database. The gene symbols and their corresponding PDB IDs representing the protein structures used in this study are human TNF-α (1TNF), human IGF1 (1B9G), human TNFR1 (1NCF), human IGF1R (7XLC), and mouse SOCS3 (2BBU, Chain A). Docking was performed at ClusPro which evaluates about 10^9^ positions of the ligand relative to the receptor. The ligand undergoes 70,000 rotations, and for each rotation, the relative three-dimensional position of the ligand with respect to the receptor was scored. Of the 70,000 rotations, the top 1000 with the lowest scores undergoes further clustering to determine cluster centers [46,47,48,49,50]. The model with the cluster center having the most neighbors within a 9-angstrom root mean square deviation (RMSD) radius was presented. From the top docked plausible interacting coordinates after docking, the most stable protein–protein interactions with lowest weighted scores were used for the purpose of assessments.

### 4.4. Analysis of Protein–Protein Interactions with PyMol

The docked pdb files from ClusPro were imported into PyMol software version 2.6.0 and cartoon models showing secondary structures of the interacting proteins were visualized. Structural configurations were set to show polar contacts (hydrogen bonds) between the chains that were represented as yellow dotted lines. A cut-off value of 3.2 was set to exclude weaker contacts between the interacting residues, and the number of bonds that were established between the chains were scored manually. Protein structures were rotated to project details on their interactions through hydrogen bonds, and images were acquired with different backgrounds [51].

## 5. Conclusions 

In the present work, we identify the specific binding sequences for SOCS3 on TNFR1. We also show that SOCS3 binds to TNFR1 but not with IGF1R. Additionally, we show that IGF1R interacts with TNFR1 and forms a hybrid heterotrimeric receptor. The topology and the confirmation shift in IGF-1 binding was observed in its interaction with the hybrid receptor. Importantly, the formation of hybrid heterotrimeric receptors could negate their interactions with corresponding ligands, TNF-α and IGF-1. This could be the underlying cause for the inhibition of SOCS3 when both TNF-α and IGF-1 are present. Finally, by binding to TNFR1, SOCS3 prevents receptor export which supports the role of SOCS3 as a negative feedback regulator.

### Limitations and Future Directions

In the present work, the pdb structure of mouse SOCS3 was used due to lack of a crystallographic structure of human SOCS3 in the RCSB database. Through sequence alignment, we identified that over 97% sequence homology exists between human and mouse SOCS3 residues. Our analysis of gene specific data from the microarray database showed a distinctive expression pattern of the assessed genes. It is not surprising that due to low sample replicates in the microarray data (*N* = 2), significant differences between the groups could not be deduced. As the data available was from technical replicates, formal statistical evaluations are not feasible. This is anticipated since the results from the microarray are often indicative and, in our experience, this is not an uncommon observation and needs additional studies. To our knowledge, our findings presented here demonstrate for the first time the formation of heterotrimeric hybrid receptors between TNFR1 and IGF1R, and their functional significance. Additional in vitro and in vivo confirmative studies such as CryoEM assessments, evaluation of receptor interactions through surface plasma resonance, and studies with TNFR1 KO mice would add more value to these findings.

## Figures and Tables

**Figure 1 ijms-26-10027-f001:**
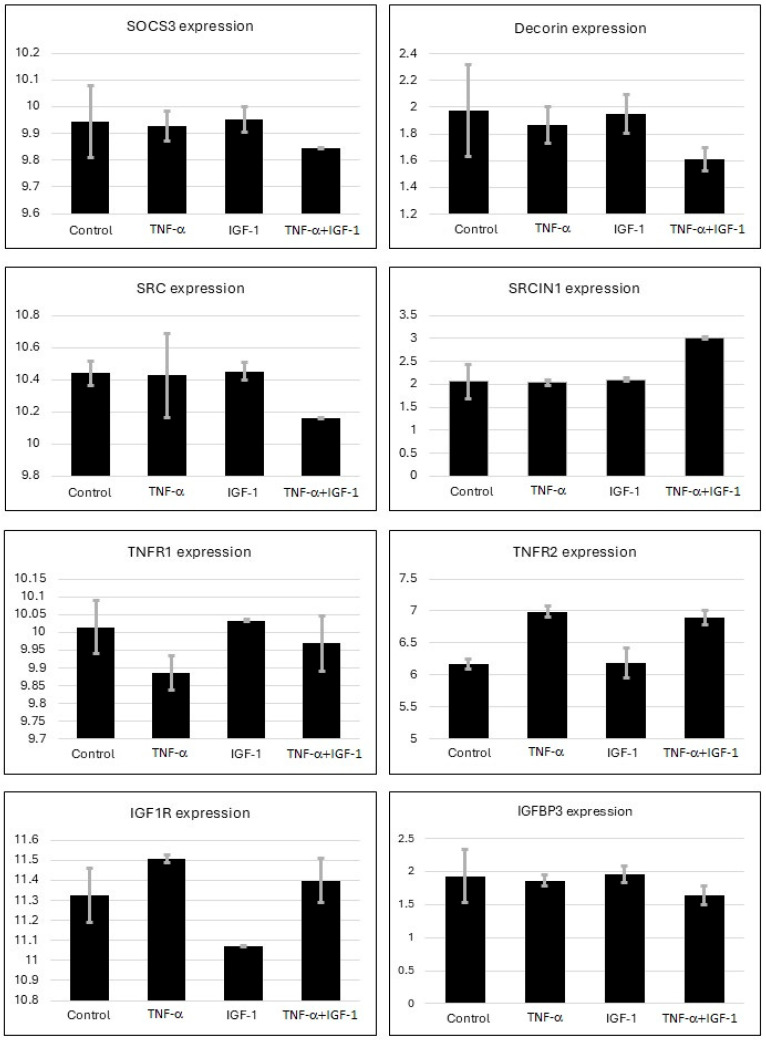
Analysis of gene expression from microarray database. Expression levels of select genes that mediate TNF-α- and IGF-1-induced pathways were assessed using GSE119651 microarray database. Figure shows expression levels of SOCS3, DCN, TNFR1, TNFR2, IGF1R, IGFBP3, SRC and SRCIN1 genes. Fold change in gene expression levels is represented on *y*-axis. Levels of gene expression with error bars representing standard error of mean from duplicate samples are shown. *p*-values could not be determined since data represents technical replicates.

**Figure 2 ijms-26-10027-f002:**
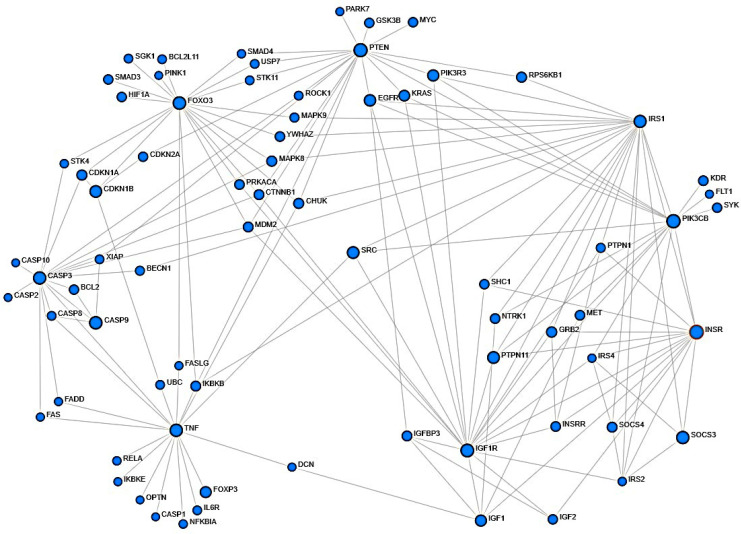
TNF-α- and IGF-1-mediated pathway mediators. Predicted pathway intermediates in TNF-α- and IGF-1-induced signaling. Selectively, proteins such as SRC, DCN, and IGFBP3 which can potentially mediate the costimulatory effects of TNF-α and IGF-1 are shown whose expression was evaluated in Figure 1.

**Figure 3 ijms-26-10027-f003:**
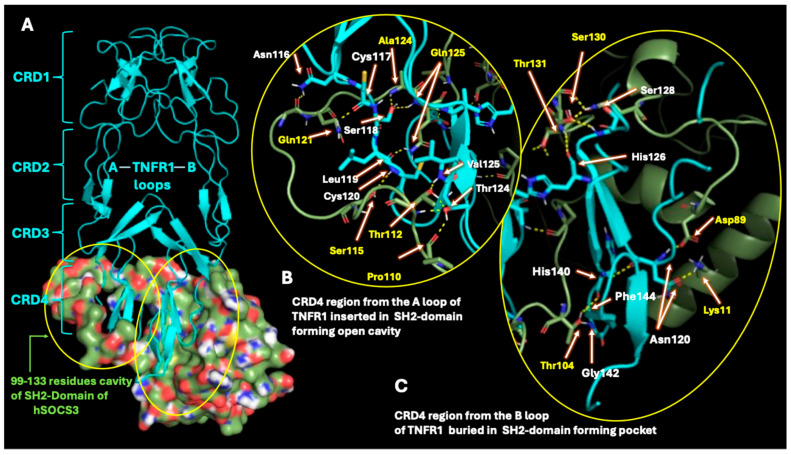
Structural insights into SOCS3–TNFR1 interactions identified by protein–protein docking analysis. (**A**) Schematic representation of TNFR1 showing cysteine-rich domains (CRD1–CRD4). The CRD4 region from TNFR1 interacts with the SH2-domain of SOCS3, with residues 99–133 forming an open cavity for docking. The regions circled with yellow line in panel A, were detailed in panels B & C, respectively. (**B**) Enlarged view of A-loop of TNFR1 CRD4 inserted into SH2-domain cavity of SOCS3. Key interacting residues include Pro110, Thr112, Ser115, Asn116, Cys117, Ser118, Leu119, Cys120, Ala124, Thr124, Val125, and Gln125 from TNFR1, which form multiple polar and hydrogen bond interactions with SOCS3 residues Pro110, Thr112, Ser115, Gln121, Ala124, and Gln125. (**C**) Enlarged view of B-loop of TNFR1 CRD4 buried within an additional pocket formed by the SH2 domain, N-terminal region, and central β-sheet residues of SOCS3. Here, TNFR1 residues Asp89, Lys11, Thr104, Asn120, His126, Ser128, Ser130, Thr131, His140, Gly142, and Phe144 establish polar contacts with SOCS3 residues Lys11, Asp89, Thr104, Ser130, and Thr131. Together, these dual binding modes highlight the structural complementarity of SOCS3 and TNFR1, suggesting a potential regulatory interface for TNF-α signaling. Figure was generated in PyMol 2.6.0 (Schrödinger) software, TNFR1 receptors are shown in cyan and SOCS3 was shown in green.

**Figure 4 ijms-26-10027-f004:**
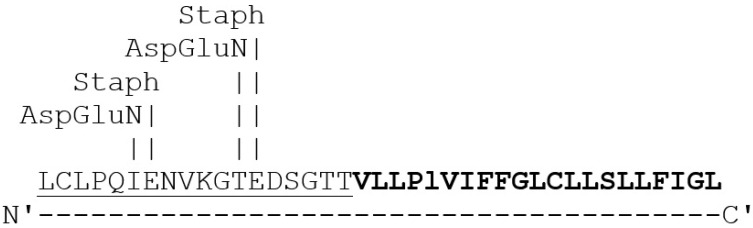
Protease cleavage sites on TNFR1: Sequence shown is in the ectodomain of TNFR1 receptor that is present N-terminal to the transmembrane domain. Sequence underlined is susceptible sequence for proteolytic cleavage by proteases from S. *aureus* (Staph) and *Pseudomonas* sps (AspGluN). Sequence highlighted in bold represents transmembrane domain of TNFR1.

**Figure 5 ijms-26-10027-f005:**
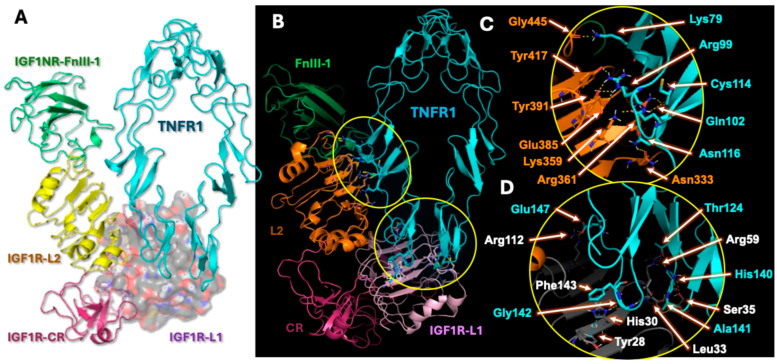
Structural modeling of IGF1R–TNFR1 hybrid receptor complex and interfacial interactions. (**A**) Docked heterotrimeric complex of IGF1R (colored domains: L1—violet, L2—orange, CR—magenta) and TNFR1 (cyan) showing overall architecture. (**B**) Ribbon representation highlighting CRD3 region of TNFR1 and L1/L2 domains of IGF1R involved in hybrid receptor formation. The regions circle with yellow line in panel B were detailed in panels C & D, respectively. (**C**) Enlarged view of the IGF1R-L2–TNFR1 interface displaying polar interactions between IGF1R residues (Gly445, Tyr417, Tyr391, Glu385, Lys359, Arg361, Asn333) and TNFR1 residues (Lys79, Arg99, Gln102, Cys114, Asn116). (**D**) Detailed view of the IGF1R-L1–TNFR1 interface showing hydrogen bonding and π–π stacking among IGF1R residues (Arg59, Arg112, His30, Tyr28, Leu33, Ser35) and TNFR1 residues (Thr124, Glu147, His140, Ala141, Gly142, Phe143). These structural insights reveal stable interdomain interactions that support hybrid receptor assembly and suggest potential overlapping ligand-binding sites for IGF-1 and TNF-α. Figure was generated in PyMol 2.6.0 (Schrödinger) software.

**Figure 6 ijms-26-10027-f006:**
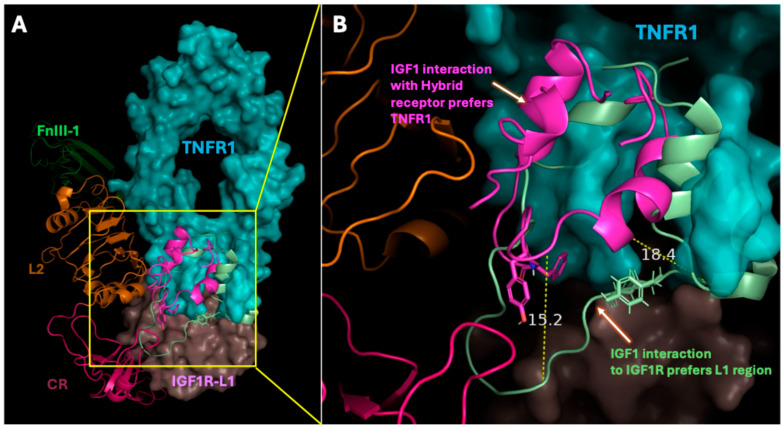
Structural insights into IGF-1 binding orientation within the IGF1R–TNFR1 hybrid receptor complex: (**A**) Overlay of IGF-1 (green ribbon) on L1 region (wheat tint) of IGF1R and IGF-1 (magenta ribbon) on CRD4 regions, loops A and B (cyan) of TNFR1. The region outlined with yellow line in panel (**A**) was detailed in panel (**B**). (**B**) Zoom view of showing overlay of IGF-1s, indicating spatial orientation of the IGF-1 on TNFR1 (cyan space) to that of IGF-1 (green ribbon) on IGF1R-L1 (wheat tint space). Figure was generated in PyMol 2.6.0 (Schrödinger) software.

**Table 1 ijms-26-10027-t001:** Analysis of structural interactions between proteins whose expression was regulated by TNF-α- and IGF-1-induced costimulatory effects. Protein–protein interactions with lowest energy scores were shown as weighted score.

Sl No	Protein 1 (PDB ID)	Protein 2 (PDB ID)	Number of Polar Contacts Between Protein 1 and Protein 2 (Members in Cluster)	Weighted Score
1.	hTNFR1 (1NCF)	hIGF1 (1B9G)	1 (86)	−848.7
2.	hTNF-a (1TNF)	hTNFR1 (1NCF)	14 (61)	−1035.6
3.	hIGF1R (7XLC)	hIGF1 (1B9G)	15 (270)	−1072.5
4.	hTNFR1 (1NCF)	hIGF1R (7XLC)	29 (47)	−1090.3
5.	hTNFR1 (1NCF)	mSOCS3 (2BBU, Ch-A)	6 (128)	−1212.0

## Data Availability

Data derived from public domain resources: The data presented in this study are available in [Gene Expression Omnibus https://www.ncbi.nlm.nih.gov/geo [reference number GSE119651]. These data were derived using the PDB ids listed above, from the following resources available in the public domain: [https://www.rcsb.org].

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
