# Peer review of "Hybrid Receptor-Mediated Molecular Delineations in TNF-α and IGF-1-Induced Costimulatory Effects"

_ijms, 2025, doi:10.3390/ijms262010027_

Round 1
Reviewer 1 Report
Comments and Suggestions for Authors
The use of transcriptomic data, protein docking, and structural analysis provides a multifaceted approach to understanding the molecular mechanisms. However, there are some missing data, which I cannot recommend the article for publication without major revisions
Major comments
- The authors should refer to in vitro experiments with some details, especially those involving cell culture and treatment, to provide complete methodological transparency.
- The authors should clearly justify their choice of BeWo cells to study TNF-α and IGF-1 signaling and describe how these cells model the inflammatory conditions of interest.
- The discussion should shed light on the potential therapeutic implications of targeting hybrid receptors in inflammatory and tumor-related diseases.
Minor comments
- In lines 27-28: either use “R” or “receptor” instead of TNFR1 receptor and IGF1R receptor
- Line 31: the sentence “prevents receptor export of TNFR1” export to where? This needs clarification
- Line 50: outside-in signaling is not a common terminology; do the authors mean upstream signaling
Author Response
Reviewer 1. Comments and Suggestions for Authors
Overall comment: The use of transcriptomic data, protein docking, and structural analysis provides a multifaceted approach to understanding the molecular mechanisms. However, there are some missing data, which I cannot recommend the article for publication without major revisions
Authors response: We appreciate the insightful comments of the reviewer. In the revised manuscript, figures 3, 5 and 6 were redone to provide more detailed view on the structural aspects of the interactions. Additional new information in Figure 6b that shows in depth details on the alterations in the binding characteristics of IGF-1 with the hybrid receptor. We also added more detailed information regarding the interactions of hybrid receptors. All changes made were highlighted in red. We believe that the revisions made based on both the reviewer’s comments have addressed these concerns.
Major comments
Comment 1: The authors should refer to in vitro experiments with some details, especially those involving cell culture and treatment, to provide complete methodological transparency.
Authors response: We apologize to the reviewer for any confusion that may have caused. We agree that the cell culture and treatment methods described in the cited publication (Tanaka R, et. al., 2018) were not detailed enough. However, we noticed that the other cited publication (Tanaka R, et. al., 2020) provides minimal information about these methods although were not detailed as identified by the reviewer. The above two publications were cited in methods for “Transcriptomic analysis”. We presume that rewording the same in the present manuscript would initiate copyright permissions, and we are certain that the reviewer would agree such processes are complicated requiring negotiations between MDPI and Springer publishing groups. To avoid such complications, we have not included those methods. However, we welcome additional specific suggestions from the reviewer in addressing this concern.
Comment 2: The authors should clearly justify their choice of BeWo cells to study TNF-α and IGF-1 signaling and describe how these cells model the inflammatory conditions of interest.
Authors response: We thank the reviewer for questioning the logical reason, which indeed has an excellent attribution to using BeWo cells. In cell culture system, the stimulating agents such as TNF-α and IGF-1, often are from extraneous sources. BeWo cells are human choriocarcinoma cell line isolated from brain metastasis tissue of a male fetal placental tumor. These are primarily epithelial like trophoblast cells. In the inflammatory disorders such as preeclampsia, the embryos in pregnancies experience stimulants that are produced from the maternal cells. Thus, the source of TNF-α and IGF-1 to embryos would be extraneous, and this attribute mimics in vitro cell culture treatments with TNF-α and IGF-1. Both TNF-α and IGF-1 play a crucial role in promoting inflammatory mechanisms during preeclampsia that affects childbirth. Thus BeWo cells appear to be well suited to study the effects of TNF-α and IGF-1 mediated signaling and cellular inflammation. As suggested, this justification was provided in the revised manuscript.
Comment 3: The discussion should shed light on the potential therapeutic implications of targeting hybrid receptors in inflammatory and tumor-related diseases.
Authors response: We thank the reviewer for this suggestion. The therapeutic implications resulting from the synergistic effects of TNF-α and IGF-1 have been reported in prominent human diseases and complications such as Duchenne muscular dystrophy, Preeclampsia, Alzheimer's disease, lung cancer, and Cardiovascular restenosis (PMIDs: 18215180, 19531281, 27344406, 11889017, 30067080). As suggested by the reviewer, we included this information in the revised manuscript.
Minor comments
Comment 1: In lines 27-28: either use “R” or “receptor” instead of TNFR1 receptor and IGF1R receptor
Authors response: Thank you for identifying this redundancy. The same was corrected in the revised manuscript.
Comment 2: Line 31: the sentence “prevents receptor export of TNFR1” export to where? This needs clarification
Authors response: We thank the reviewer for noting this and apologize for the missing information. We were referring to the “export of TNFR1 to cell surface”. We made this suggested change in the revised manuscript.
Comment 3: Line 50: outside-in signaling is not a common terminology; do the authors mean upstream signaling
Authors response: This is an interesting question. We and others have previously used this terminology “outside-in signaling” to describe integrin mediated signaling, which are cell surface receptors. We noticed frequent use of this terminology in recent publications on GPCR and other receptor mediated signaling (PMIDs: 39905041, 26345366). We also agree with the reviewer that it would indicate upstream signaling, but more precisely as a receptor mediated since upstream signaling can be intracellular as well.
We hope to have addressed this reviewers' comments satisfactorily. Furthermore, we welcome any additional comments which the reviewer may have during the re-review process. Thank you.
Reviewer 2 Report
Comments and Suggestions for Authors
The manuscript presents an integrative analysis combining gene expression profiling with structural docking simulations to explore the synergistic effects of TNF-a and IGF-1 signaling. The study is of potential interest, however, the presentation requires significant improvements in clarity, methodological rigor, and biological interpretation. The results are described in detail, but some sections are difficult to follow due to long sentences and redundancy. Moreover, statistical considerations and functional implications should be more clearly addressed.
Major Comments
- The analysis is based on duplicate samples. Please clarify whether these are technical or biological replicates. With such a small sample size, statistical significance is limited; indicate how this limitation was handled
- While structural interactions (SOCS3–TNFR1/2, hybrid IGF1R:TNFR1 complexes) are well described, their functional implications remain underexplored. What do these interactions mean for signaling outcomes? Are they inhibitory, synergistic, or competitive?
- Some results are referenced as “data not shown.” Please either include the missing data or justify their exclusion.
Author Response
Reviewer 2. Comments and Suggestions for Authors
Overall comment: The manuscript presents an integrative analysis combining gene expression profiling with structural docking simulations to explore the synergistic effects of TNF-a and IGF-1 signaling. The study is of potential interest, however, the presentation requires significant improvements in clarity, methodological rigor, and biological interpretation. The results are described in detail, but some sections are difficult to follow due to long sentences and redundancy. Moreover, statistical considerations and functional implications should be more clearly addressed.
Authors response: We commend reviewers’ favorable comments here. As suggested, we revised the manuscript and have also addressed the below concerns. The revised manuscript shows changes made based on the suggestions from both the reviewers. The changes were highlighted in red, in the revised manuscript.
Major Comments
Comment 1: The analysis is based on duplicate samples. Please clarify whether these are technical or biological replicates. With such a small sample size, statistical significance is limited; indicate how this limitation was handled
Authors response: The samples are technical replicates since the in vitro cell culture treatments were analyzed in duplicate using BeWo commercial cell line. As identified by the reviewer, we agree that due to the small sample size in the microarray data, the statistical significance was limited. Formal statistical testing is not suitable as the available data represent only technical replicate groups tested rather than independent biological replicates. Thus, our focus has been on the observable trends in gene expression, which is valuable in validating our hypothesis. Importantly, this external data from NCBI database was generated without prior knowledge of our hypothesis, thereby eliminating the bias. We attempted to conduct statistical evaluations stated below and excluded them based on the nature of the experimental data. 1) Student's t-test (considering equal variances) and 2) Welch's t-test (assuming unequal variances due to unanticipated effects). 3) Mann-Whitney U test (values tied to the average ranks) along with the inclusion of appropriate correction parameters as feasible. 4) The non-parametric Wilcoxon signed-rank test and 5) Turkey test (based on ANOVA from four groups). These tests were deemed not suitable for analysis and is anticipated because the microarray data would be mostly indicative when samples were only run in duplicates. Such observations warrant additional evaluations and statistical analysis using gene specific primers. On that note, in our previous publications that were cited, we conducted statistical tests using RT-PCR of data from select gene specific primers, which showed significant differences between the groups (PMIDs: 26047583, 23335796, 21540027). In the present manuscript, although markings for data comparison were not shown in Fig.1 due to lack of significance, the p-values from Students t-test (equal variance was anticipated as the samples were technical replicates) were considered and were not significant. We agree with the reviewer on the sample size and nature of the microarray data, and we welcome additional specific suggestions from the reviewer in analyzing this data to present it in a more meaningful way when recommended.
Comment 2: While structural interactions (SOCS3–TNFR1/2, hybrid IGF1R:TNFR1 complexes) are well described, their functional implications remain underexplored. What do these interactions mean for signaling outcomes? Are they inhibitory, synergistic, or competitive?
Authors response: We agree with the reviewer, and support that additional studies are needed to decipher functional significance stemming from the interactions between these receptors, and we proposed a few of these studies in the manuscript which are not feasible to conduct at this time. Based on the structural interactions, we anticipate that the formation of IGF1R:TNFR1 hybrid receptor would prevent downstream receptor mediated signaling from both the receptors. Thus, the outcome could have an inhibitory effect. Data from Fig.1 also indicates that TNFR2 would play a compensatory role when TNFR1 is not active. Since the role of TNFR2 is not as robust as TNFR1, we speculate that the compensatory role of TNFR2 is required for cell survival. We previously observed such inter-regulatory pattern between DNMT1 and DNMT3a whose expression was also regulated by TNF-α and IGF-1 (PMID: 30067080).
Comment 3: Some results are referenced as “data not shown.” Please either include the missing data or justify their exclusion.
Authors response: We appreciate the reviewer for highlighting this aspect and agree that in two instances, we mentioned as “data not shown”. In the first instance, we were describing the interaction between SOCS3 and TNFR2 receptor. Since TNFR2 is not participating in the formation of hybrid receptors with IGF-1, we did not present that information. As TNFR1 is the major receptor mediating most of the inflammatory mechanisms, we specifically investigated the properties of TNFR1. In the second instance, we mentioned “data not shown” where the interaction between IGF1R and thyrotropin receptors was discussed. This was mainly because it was a previously published work and the authors have already claimed that in their publication.
We hope to have addressed this reviewers comments satisfactorily. Furthermore, we welcome any additional comments which the reviewer may have during the re-review process. Thank you.
Round 2
Reviewer 1 Report
Comments and Suggestions for Authors
Thank you for incorporating the suggested information into the revised manuscript. The added explanation in the Methods section now makes the analytical approach much clearer.
All my previous comments have been satisfactorily addressed. I have no further concerns.
Good luck.
Reviewer 2 Report
Comments and Suggestions for Authors
The authors have addressed the issues I had highlighted. Recommendation: Accept after minor editorial revisions.